# Wafer Level Vacuum Packaging of MEMS-Based Uncooled Infrared Sensors

**DOI:** 10.3390/mi15080935

**Published:** 2024-07-23

**Authors:** Gulsah Demirhan Aydin, Orhan Sevket Akar, Tayfun Akin

**Affiliations:** 1METU MEMS Centre, Middle East Technical University, Ankara 06530, Turkey; 2Electrical and Electronics Engineering Department, Baskent University, Ankara 06790, Turkey; 3MikroSens Elektronik San ve Tic A.S., Ankara 06530, Turkey; 4Electrical and Electronics Engineering Department, Middle East Technical University, Ankara 06800, Turkey

**Keywords:** wafer level vacuum packaging (WLVP), infrared detector packaging, hermetic encapsulation, TLP bonding, glass frit bonding, grating structures

## Abstract

This paper introduces a cost-effective, high-performance approach to achieving wafer level vacuum packaging (WLVP) for MEMS-based uncooled infrared sensors. Reliable and hermetic packages for MEMS devices are achieved using a cap wafer that is formed using two silicon wafers, where one wafer has precise grating/moth-eye structures on both sides of a double-sided polished wafer for improved transmission of over 80% in the long-wave infrared (LWIR) wavelength region without the need for an AR coating, while the other wafer is used to form a cavity. The two wafers are bonded using Au-In transient liquid phase (TLP) bonding at low temperature to form the cap wafer, which is then bondelectrical and Electronics d to the sensor wafer using glass frit bonding at high temperature to activate the getter inside the cavity region. The bond quality is assessed using three methods, including He-leak tests, cap deflection, and Pirani vacuum gauges. Hermeticity is confirmed through He-leak tests according to MIL-STD 883, yielding values as low as 0.1 × 10^−9^ atm·cc/s. The average shear strength is measured as 23.38 MPa. The package pressure varies from 133–533 Pa without the getter usage to as low as 0.13 Pa with the getter usage.

## 1. Introduction

The evolution of the Micro-Electro-Mechanical Systems (MEMS) technology has rapidly advanced since the 1990s, impacting various aspects of our daily lives, especially in response to the escalating demand for intelligent systems in various applications, from automotive and smartphones to IoT devices, robotics, drones, night vision equipment, smart cities, and autonomous vehicles. A critical step on the road to the commercialization of MEMS devices is their packaging in terms of reliability and cost. While MEMS packaging is application specific, the MEMS industry is actively seeking universal and more economical approaches for packaging sensor devices while overcoming the sensor-specific challenges. Although certain MEMS sensors necessitate direct physical contact with the external environment (like in gas flow sensors and pressure sensors), the majority of MEMS sensors must be effectively isolated from the surrounding atmosphere and be in a vacuum environment to ensure proper functionality, like in inertial sensors and infrared detectors [1,2,3,4,5].

The traditional method of vacuum packaging MEMS devices involves dicing and singulating individual sensor chips from a processed sensor wafer and then encapsulating each of them in individual packages in vacuum; however, this is not only a very costly method but also results in large package sizes. An alternative to this approach is to enclose each MEMS device or component in a sealed vacuum environment in a batch fashion using wafer level vacuum packaging (WLVP), where each MEMS device in the sensor wafer is individually sealed using a bonding process with a cavity-formed cap wafer, after which the vacuum-sealed sensors are singulated using dicing. The advantages of WLVP can be listed as cost-effectiveness, size reduction, improved performance, enhanced reliability, and easy integration with other technologies.

The WLVP can be achieved through various methods, such as anodic bonding, fusion bonding, glass frit bonding, or eutectic bonding, and there are already well-established studies in the literature. Table 1 provides an overview of these studies categorized according to the bonding process, which is distinguished by factors such as process temperature, tolerance for surface irregularities, and the vacuum level. The thin-film packaging method can be used for post-CMOS processed sensors in a CMOS wafer but has limitations in achieving a high vacuum level, as it is difficult to place and activate a getter in a small volume of cavity. The anodic bonding approach is very useful for the fabrication of MEMS structures and can provide hermetic sealing, but its use in the WLVP of post-CMOS processed sensors is not convenient due to the need for a planar surface on top of feedthroughs, for a silicon layer on top of the sealing region, and for precautions to prevent possible damage during the high-voltage applied bonding step. Metal thermocompression, solder, TLP, and eutectic bonding have significant advantages, like having lower bonding temperature and good hermeticity; however, for these methods, sealing over electrically conductive feedthroughs introduces complexity in post-CMOS processed sensors for ensuring electrical isolation between the feedthroughs and the sealing rings. Additionally, the combined thicknesses of layers typically extend to 8–10 µm in the case of metal-based bonding approaches, which leads to electroplating becoming the sole practical deposition technique, as described in the literature [6,7]. Glass-frit bonding is one of the most reliable and high-yield wafer level vacuum packaging approaches applied to a wide range of sensors, including post-CMOS processed sensors. The advantages of this approach can be listed as elimination of a photolithography step for deposition and patterning, elimination of the need for a smooth surface topography on top of the feedthroughs in the sensor wafer, elimination of the need for metal deposition and patterning on the sensor wafer, as well as high hermeticity and proper activation of the getter during the high bonding temperature. It should be noted that glass frit can only be used with post-CMOS processed sensors if the sensor properties do not degrade with the high bonding temperature.

One of the most popular post-CMOS processed sensor types is uncooled thermal infrared sensors that work in the Long-Wave Infrared (LWIR) wavelength region [38]. MEMS-based uncooled sensors are widely used in various military and commercial applications, including night vision, mine detection, reconnaissance, firefighting, medical imaging, industrial control, smart buildings, smart offices, and smart agriculture [39]. These applications are pushing for monolithically fabricated post-CMOS processed uncooled sensors with a compact and low-cost package, which is making WLVP an important requirement. The selection of the WLVP approach is very critical for satisfying proper vacuum levels and for allowing infrared signals to reach the sealed sensor without too much loss.

This article introduces a versatile, high-performance WLVP solution for a CMOS-compatible uncooled infrared sensor. The experimentally verified approach involves the creation of moth-eye structures on both sides of a silicon cap wafer, eliminating the need for anti-reflective (AR) coatings. This silicon cap wafer is then bonded to another spacer wafer through a low-temperature Au-In TLP bonding process, where cavities are formed using the deep reactive ion etching (DRIE) process on the spacer wafer that incorporated the moth-eye structures within. A glass frit bonding layer is then deposited on the spacer wafer side of the cap wafer stack, which is preferred as it eliminates the need for another layer deposition on the CMOS wafer side. Finally, this cap wafer stack is bonded to a post-CMOS processed sensor wafer, including microbolometer-based Pirani vacuum gauge pixel structures, for wafer level vacuum encapsulation of the individual sensor dies. The long-term quality, reliability, and repeatability of the proposed cap wafer technology are assessed by bonding the cap wafer stack to various silicon sensor wafers and measuring bonding quality, including the vacuum level of the hermetically sealed cavity regions.

## 2. Experimental

### 2.1. Proposed Cap Wafer Fabrication Approach

Silicon wafers are preferred as the cap wafers in the low-cost option for WLVP of uncooled infrared sensors, but they reflect a significant amount of infrared radiation, reducing the transmission to about 50% in the LWIR region. The use of anti-reflective coatings on both sides of double-side polished silicon wafers is a good option for increasing the transmission to over 85%; however, the AR coating limits the bonding and getter activation temperature as it might peel off from the surface. An alternative approach is to implement moth-eye structures on the silicon surface as a solution for enhancing transmission of incoming IR radiation, which can be seen in the literature and the industry [6,40,41,42,43]. Optimization studies of anti-reflective grove-type gratings have been continuing in our research group since 2012, and the numerical modeling and simulations for pillar- and groove-type gratings with various topological configurations changing in various period sizes and heights/depths, and all these results were experimentally verified and published in [44]. However, implementation of these structures within a cavity poses significant challenges. In order to overcome these challenges, this study proposes to use double-side moth-eye structures to be implemented on a double-side polished flat window wafer, while the cavity is to be formed with a spacer wafer attached to it by using the Au-In TLP bonding technique. Although there are other material alloy options (such as Cu-Sn or Au-Sn), Au-In is selected because the Au-In material system is not only promising for allowing bonding at low temperatures (around 200 °C) and capable of withstanding higher temperatures (up to 500 °C) that are required in the following high-temperature glass frit bonding steps but also provides strong and hermetic bonds in the cavity regions of the spacer wafer formed by DRIE etching. Following the cavity formation, a lead-based glass frit paste is applied to the cap wafer stack using screen printing from the cavity side. The above cap wafer process sequence finishes with thin-film getter formation inside the cavity of each die using a shadow mask. The performance of the proposed cap wafer can be tested on microbolometer-based Pirani vacuum gauge pixel structures formed in the CMOS wafer or a dedicated sensor wafer.

After completing the fabrication of the cap and sensor wafers separately, they are bonded together using the glass frit bonding technique in a vacuum chamber for the formation of the WLVP sensor structure. Following this, the WLVP sensor wafer stack has been shallow diced first to remove the cap wafer pieces over the wire-bonding pad regions of the sensor wafer, and then the whole wafer has been diced for the singulation of the individual WLVP sensors. Figure 1 shows the detailed cross section of the proposed wafer level hermetic encapsulation method.

### 2.2. Fabrication Processes

The proposed hermetic packaging method of the IR-based thermal sensor consisted of fabrication steps of the cap wafer stack and a sensor wafer. Figure 2 presents the process flow of the cap wafer stack. The fabrication process starts with a blank 8″ double-side polished (DSP) silicon wafer as the window wafer. Both sides of the window wafer are prepared for high-resolution stepper lithography to determine the precise moth-eye structures, specifically at the regions corresponding to the IR sensor regions of the sensor wafer. The second step is the reactive ion etching (RIE) process of the window silicon wafer for the formation of the grating structures on both sides of the wafer, as shown in Figure 2a. The structures in the window wafer are protected with SiO_2_, which will also act as an etch stop layer for the following process steps, as shown in Figure 2b. The third step is the metal layer formation on the bottom side of the window wafer, which is achieved with sputtering of Ti(20 nm)/Ni(50 nm)/Au(1 µm) layers that are patterned using the lift-off technique. The spacer wafer is used for cavity formation above the sensor structure area over the sensor wafer. The spacer wafer can be either a standard thickness (approximately 725 μm) SSP Si wafer that can be grinded down to 400 µm or below in the following steps or a 400 μm thin Si DSP wafer that eliminates the grinding step. Figure 2c shows the metal layer formation on the spacer wafers, which is achieved with sputtering of Ti(20 nm)/Ni(50 nm)/Au(500 nm) layers and then evaporation of In(3 µm), which are patterned with the lift-off technique. After metal patterning on both wafers, the two wafers are bonded together using a 30 min Au-In TLP bonding process in which a bond pressure of around 2.2 MPa is applied, with a low bonding temperature of 200 °C while the chamber vacuum level is 0.001 Pa (10^−5^ mbar); then, the wafer stack becomes a single part, as shown in Figure 2d. When a standard thick SSP silicon spacer wafer is used, the spacer wafer can be grinded down to the desired spacer thickness after the wafer bonding. The cap wafer fabrication continues with the cavity lithography on the bottom side of the cap wafer stack. Depending on the spacer wafer thickness used, 200–400 μm deep cavities have been etched on the cap wafer stacks to obtain a vacuum cavity after the bonding, where precise predefined grating structures are revealed. At the end of the cavity formation, the DRIE etch stop oxide layer and the protection layers are removed from the window wafer, as depicted in Figure 2e. Then, the glass frit, including a 200 µm bond frame width, is deposited using the screen-printing method on the bottom side of the cap wafer stack, as shown in Figure 2f. Following the thermal conditioning (frit firing) processes, a getter material is sputtered inside the cavity region using a shadow mask. The getter deposition finalizes the cap wafer stack fabrication, and the wafer is ready for glass frit bonding with the sensor wafer, as shown in Figure 2g.

A sensor wafer is also designed and fabricated to be used for the characterization and demonstration of the proposed WLVP technique. Figure 3 shows the fabrication steps of the sensor wafer, which consists of the microbolometer-based Pirani vacuum gauge pixel structures and metal routing lines on each die area. First, a silicon nitride dielectric layer is deposited on the front side of a regular 8″ silicon wafer for electrical isolation, and then Ti/Au metal layers are deposited and patterned with wet etching for the formation of metal routing lines and the wire-bonding pads (Figure 3b). A sacrificial layer is then coated and patterned with the anchor points of the pixel structure (Figure 3c); the structural nitride layer is deposited on top of the sacrificial layer, and via openings are etched at the contact points (Figure 3d). The interconnect metals are deposited and patterned on the support arms (Figure 3e). The active material is then deposited and patterned on the pixel body to obtain a high temperature coefficient of resistance (TCR) value (Figure 3f). Finally, the passivation nitride layer is deposited and patterned with plasma etching to form the pixel structure (Figure 3g). After the formation of the microbolometer-based Pirani vacuum gauge structures, the sacrificial layer is removed for the suspension of the pixel structures. At the end of this process step, the sensor wafer fabrication is completed and ready for cap wafer bonding for the WLVP (Figure 3h).

The prepared cap wafer stack and sensor wafer are aligned and bonded using the glass frit bonding process to form the wafer level vacuum packaged sensors and verification of the proposed WLVP approach. Wafer level hermetic packaging is achieved by the optimized glass frit bonding technique, in which the bonding occurs at around 430 °C for 20 min by using a bond pressure of around 3 MPa, considering the bonding area over the wafer, while the chamber vacuum level is 0.01 Pa (10^−4^ mbar); and the final glass frit bond frame width is measured as 400 µm after the bonding. The cap wafer is then diced partially to access the wire-bonding pads of the sensors. Figure 4a shows the perspective view of the actual fabricated wafer stack, while Figure 4b shows the picture of the packaged sensor after the singulation of the individual wafer level packaged sensors. Figure 5 shows the SEM images of the WLVP MEMS-based sensor, including (a) the details of the encapsulated MEMS sensor, (b) Au-In TLP and glass frit bonding regions, (c) the cavity region and gratings (moth-eye structures) inside the cavity, and (d) a perspective view of the bonding pad region of the sensor chip after cap wafer pad reveal.

## 3. Results & Discussion

### 3.1. Transmission Behavior of Double-Side Anti-Reflection Moth Eye (Grating) Structures

Infrared transmission measurements are performed on the wafers using an IR-VASE ellipsometer system to compare the infrared transmission characteristics of a non-processed bare silicon wafer and a double-side anti-reflective moth-eye structure integrated silicon wafer. Figure 6 shows the IR transmission characteristics of the measured wafers in the 2–20 μm wavelength region. The integration of double-side moth-eye structures on the window wafer enhanced the IR transmission from approximately 50% up to 82% in the 8–12 μm wavelength region.

### 3.2. Au-In TLP Bonding Quality

The bonding quality of the bonded wafer stack is inspected using a scanning acoustic microscope (SAM) system, as shown in Figure 7a. Figure 7b shows the SAM image of the Au-In TLP bonded cap wafer stack, and Figure 7c shows a closer view of the eight-die area. The dark colors refer to the continuous penetration of ultrasound waves in the scanned image. The bond ring regions are in dark gray color, which refers to a continuous material structure in the bond ring region. The leaked water in the streets between dies can be seen in the image as a light gray color, and finally, the other light side means gratings and unbonded parts. The SAM image also shows some defective regions, which may have resulted during the microfabrication process or handling. After bonding, squeeze-out was observed around the bond rings, and the outer circular part of the 8″ wafer refers to the reflow of the melted indium metal that occurs during the bonding process, which was good evidence for high-quality bonding, as liquefaction was clearly detected.

The mechanical strength of the Au-In TLP bonding interface is analyzed by applying shear tests on the individually diced nine samples from different locations on the monitoring wafer, and the average strength is measured to be 32.10 ± 5.29 MPa, as shown in Figure 8. The measured value satisfies the military standard MIL-STD 883E [45], where the minimum shear strength value is 6 MPa for microelectronic packages. Seven additional samples are also tested from monitoring wafer after a year, it can be pointed out that the TLP bonding remains strong, as shown in red color dots in Figure 8.

After conducting shear tests, the squeezed-out indium can be seen clearly in Figure 9a. A narrow region, measuring only a few micrometers wide, appears on the shear surface, contrasting with a highly deformed area elsewhere in the bond ring. This deformation resembles ductile fracture, evidenced by dimples and cones. Elemental analysis via Energy Dispersive Spectroscopy (EDS) of the narrow region containing bulk bonding material showed a composition of 42 wt. % In, closely matching the targeted total composition. In contrast, EDS analysis of the deformed region indicated a composition of 77.4 wt. % Au (Figure 9b). The failure of the bond has likely occurred from the AuIn intermetallic and a closure view for the SEM image of the shear surface can be seen in Figure 9c.

Evaluation of the bonding quality at various temperatures is continued with the mechanical robustness tests, including thermal cycling (room temperature to 250 °C) for 5 cycles, high-temperature storage (300 °C for a day), and ultra-high-temperature shock tests according to MIL-STD-810 criteria [46]. Higher temperature tests are also conducted to see if the Au-In TLP bonded lid withstands the following glass frit bonding and getter activation steps, which are to be performed at around 430 °C. Three different test groups are arranged, including annealing at 430 °C for 10 min, 30 min, and 60 min. After each annealing step, destructive shear tests are applied. The average shear strength is measured as 26.41 ± 10.14 MPa after 10 min of annealing. After 30 min of annealing, the average shear strength is measured as 28.90 ± 5.91 MPa. For the 60 min annealing case, the average shear strength is measured as 27.61 ± 10.25 MPa. Figure 10 shows the before and after annealing shear test results plotted in the same graph for 4 different regions of the wafer for comparison, where the average shear strength results are presented for each region. A total of 44 dies are shear tested in the four regions; during the tests, 4 dies fail, and in some dies, weak results are obtained due to the lack of poor bonding caused by voids or dust. These results verify that all regions satisfy the MIL-STD 883 criteria. 

### 3.3. Glass Frit Bonding Quality

The bonding quality and strength of the glass frit bonding are also inspected on one of the sensor wafers bonded to the cap wafer. Figure 11 presents the result of the high-resolution scanning acoustic microscope mapping, showing uniform bonding results of the stack, where the enlarged picture clearly shows different regions of the bonding area.

The shear strength of 44 packaged samples is measured from different locations on the wafer to quantify the bonding strength and long-term reliability of the glass frit bonding process. The average shear strength is initially measured as 27.04 ± 6.45 MPa, which is very good, as expected.

### 3.4. Hermeticity Tests of the Proposed Cap Wafer Fabrication Method

The hermeticity of the proposed cap wafer fabrication method is also tested after wafer level packaging with the sensor wafer (a total of 2 microbolometer-based Pirani vacuum gauge sensor wafers are fabricated) and without (a total of 8 wafers) Pirani vacuum sensors. The three different methods used are He-leak test, cap deflection measurement, and using a microbolometer-type vacuum sensor packaged with the proposed cap wafer.

The hermeticity monitoring of the samples is characterized with He-leak tests. A 5 × 7 sensor block is used during the tests for easy measurement, since one die cavity volume is very small (7.28 mm^3^) according to MIL-STD-883 package volumes and defined requirements. The 5 × 7 sensor block piece is exposed to helium bombardment by using a pressure vessel at 75 psi for 16 h according to the defined leak test parameters in MIL-STD-883E. Then, the sensor block is placed in a He-leak test chamber, the chamber is pumped down, and the vacuum pump exhaust is measured with a He detector, obtaining a 0.1 × 10^−9^ atm·cc/s leak rate. This measurement verifies that the rejection leak rate limit of 5 × 10^−8^ atm·cc/s specified in the MIL-STD 883 criteria is satisfied on the tested 5 × 7 sensor block.

The vacuum level inside the cavity can also be calculated by measuring the cap diaphragm deflection after the diaphragm thickness is reduced to make the cap diaphragm thin, allowing it to be deflected by the small cavity vacuum. The diaphragm thickness reduction of the abovementioned 5 × 7 sensor block piece is achieved by etching the top of the window wafer using a DRIE system. Equation (1) can be used to calculate the cap deflection and corresponding vacuum level with the dimensions and mechanical properties of the cap diaphragm given in Table 2,
(1)$$∆P_{total}=\frac{32Eh^{3}\left(l^{4}+w^{4}\right)}{5(1-\nu^{2})l^{4}w^{4}}∆\omega_{max}$$
where $P_{total}$ is the total pressure, E is the Young’s modulus of silicon, h is the diaphragm thickness, ν is the silicon Poisson’s ratio, $\omega_{max}$ is the maximum deflection, and finally l and w are the length and width of the diaphragm, respectively.

Figure 12a shows the picture of the thinned 5 × 7 sensor block where the diaphragm deflections can be seen as curved small rectangles on the surface, while Figure 12b shows the diaphragm deflection measured in both 2D and 3D surface profilers as 16.8−20.1 µm at the middle dies and 26.7–37.9 µm at the edge dies. The variation in the center deflection is due to the non-uniform etching of the sample block during the DRIE etching step; the edge region is etched more compared to the center region, as can be seen from the deflection amounts. After these measurements, the diaphragm of the most deflecting die, i.e., the thinnest die, is intentionally broken to measure the diaphragm thicknesses with cross-sectional SEM inspections; this diaphragm is measured to be 30 µm thick. This result is consistent with the COMSOL diaphragm deflection simulation results provided in Figure 12c,d. An about 39 µm central deflection is also achieved with a 30 µm thick diaphragm, suggesting a cavity vacuum level of approximately 500 Pa (5 mbar).

Another method for monitoring the vacuum level inside the package is to use microbolometer-based Pirani vacuum gauge pixel structures inside the capped region, as shown in Figure 3. The Pirani vacuum gauge pixels are first characterized in a vacuum probe station before cap wafer bonding and measured again after cap wafer bonding. The first measurement results are used as the lookup table for determining the vacuum level inside the packaged area.

As the first step of the characterization, functional sensors are selected by using a current source meter and applying a current from 0 nA to 3000 nA with 150 nA increments at 1 atm, 101,325 Pa pressure level at 25 °C chuck temperature. The TCR of a material is defined as in Equation (2):(2)$$\alpha =\frac{1}{R}\frac{dR}{dT}\;\;$$
where α is the temperature coefficient of resistance, R is the total resistance of the material, and T is the temperature. The measurement of TCR is performed in the probe station system’s temperature-controllable chuck and chamber by using an unsuspended detector to eliminate the effect of electrical heating. The temperature of the environmental chamber is changed from 22 °C to 38 °C, and the voltage of the sensor is measured while it is biased with a constant current (300 nA). The temperature data are obtained from the probe system’s chuck heater temperature controller unit, and the total resistance values changing by the temperature are plotted in a graph and fitted to a polynomial. The TCR value at room temperature is measured as approximately −5.5%/K.

The wafer level sacrificially released sensors are then tested for thermal conductance measurements in a vacuum probe station system, and I-V sweeps are performed at predefined vacuum levels of 0.13 Pa, 2.67 Pa, 6.67 Pa, 13.33 Pa, 19.99 Pa, and 26.66 Pa, respectively. The thermal conductance versus pressure plot for all the functional dies is plotted by using the formula in Equation (3) [38] to obtain a lookup table to be referenced after wafer level vacuum packaging.
(3)$$G_{th}=\frac{P_{elec}}{\Delta T}=\frac{I^{2}R\alpha }{ln(\frac{R}{R_{0}})\;}\;\;\;$$
where G_th_ is the thermal conductance of the detector, *P_elec_* is the applied electrical power, Δ*T* is the total temperature change, *R* is the detector resistance under vacuum, *R*_0_ is the detector resistance at atmospheric pressure, *α* is the TCR of the detector resistance, and *I* is the applied bias current.

The wafer is tested again after wafer level vacuum packaging, and the vacuum level inside the package is estimated according to the obtained lookup tables and plots by using the calculated G_th_ value. It should be noted that, different from cap deflection characterization, the sensor wafer has a getter to evaluate the increase of the vacuum level inside the package, and the pressure inside the package is measured to be in the range of 133–533 Pa if the getter is not used, whereas it is 0.13–13.33 Pa with the getter usage.

These test and characterization results verify that the proposed cap wafer fabrication method is successfully processed in 8″ wafers. Table 3 summarizes the processes and pressure ranges of the packaged Pirani gauges in the literature and those used in this work.

## 4. Conclusions

This work presents an alternative approach to obtaining wafer level vacuum packaging that satisfies the requirements of CMOS-based thermal sensors at low cost and with high performance. The proposed method is experimentally verified by fabricating the moth-eye structures on an 8″ DSP wafer and demonstrating IR transmission performance of about 80%, eliminating the need for high-cost AR coating; by optimizing and verifying the TLP bonding performance of the window cap and spacer wafers; by optimizing the cavity opening step without damaging the moth-eye structures; by optimization of glass frit deposition and glass frit bonding for 8″ size wafers; by developing a vacuum sensor wafer with Pirani vacuum gauges for bonding quality measurements; by bonding the cap wafer stack to various silicon wafers; and by measuring bonding quality, including the vacuum level of the hermetically sealed cavity regions. The best package pressure is measured to be around 133–533 Pa if a getter is not used, whereas the best pressures are measured to be 0.13 Pa in the case of getter usage.

## Figures and Tables

**Figure 1 micromachines-15-00935-f001:**
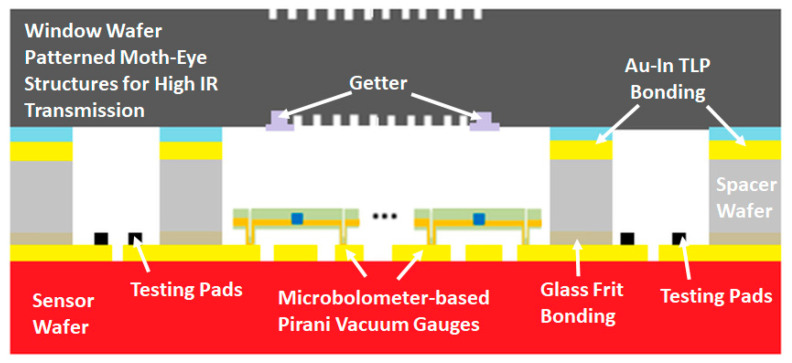
The cross section of a microbolometer-based Pirani vacuum gauge sensor die area that is wafer level hermetically encapsulated on an 8″ wafer for MEMS-based LWIR sensors.

**Figure 2 micromachines-15-00935-f002:**
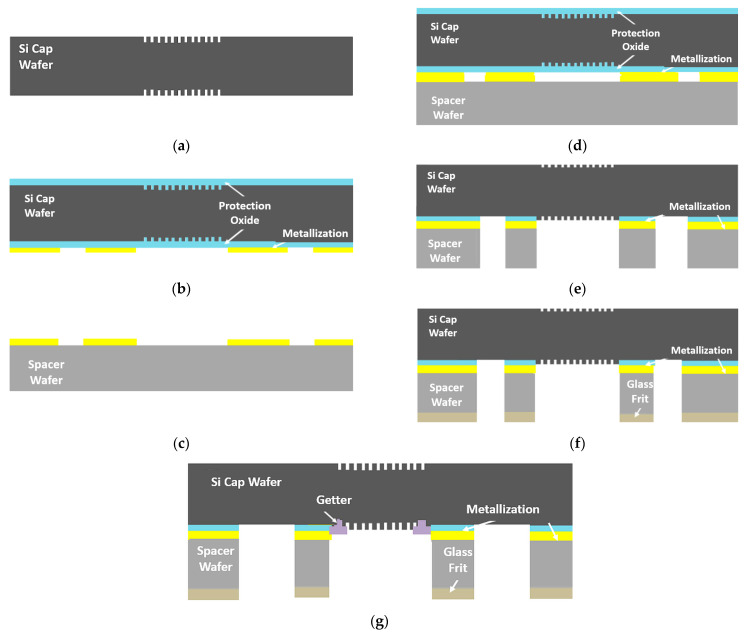
Schematic of the proposed cap wafer process flow: (**a**) formation of moth-eye structures on both sides of the window wafer; (**b**) coating of protection oxide and metal formation; (**c**) spacer wafer metallization; (**d**) Au-In TLP bonding of the wafers; (**e**) DRIE etching for cavity formation and protection oxide etching; (**f**) glass frit paste deposition; (**g**) thin-film getter deposition.

**Figure 3 micromachines-15-00935-f003:**
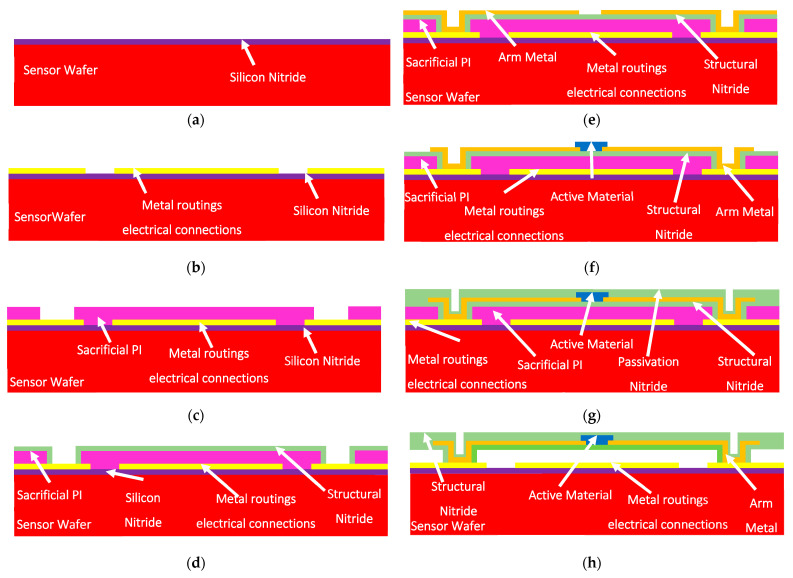
Schematics of the sensor wafer process flow: (**a**) start with a DSP silicon wafer; (**b**) formation of metal routings and electrical connections; (**c**) deposition and patterning of the sacrificial layer; (**d**) deposition of the structural nitride layer and opening of the contacts at the anchor points; (**e**) deposition and patterning of the interconnect metal on the support arms; (**f**) deposition and patterning of the active material; (**g**) passivation nitride layer deposition and plasma etching to form the pixel structure; (**h**) the sacrificial PI layer removal for the formation of the thermally isolated microbolometer-based Pirani vacuum gauge pixel structure.

**Figure 4 micromachines-15-00935-f004:**
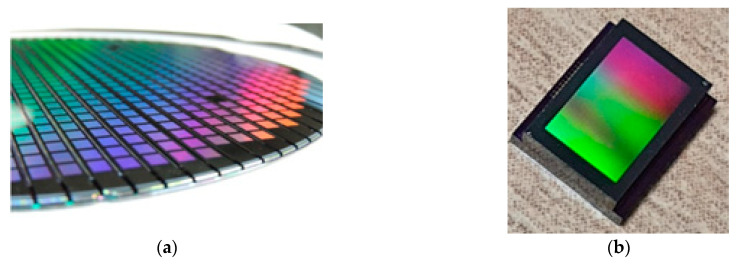
Pictures of the fabricated IR-based thermal sensors after WLVP with the proposed cap wafer fabrication technique: (**a**) the picture of the wafer level vacuum packaged sensor wafer after shallow dicing for pad reveal; and (**b**) the picture of one of the sensor dies (dimensions are 5.4 × 6.5 mm^2^) after singulation dicing.

**Figure 5 micromachines-15-00935-f005:**
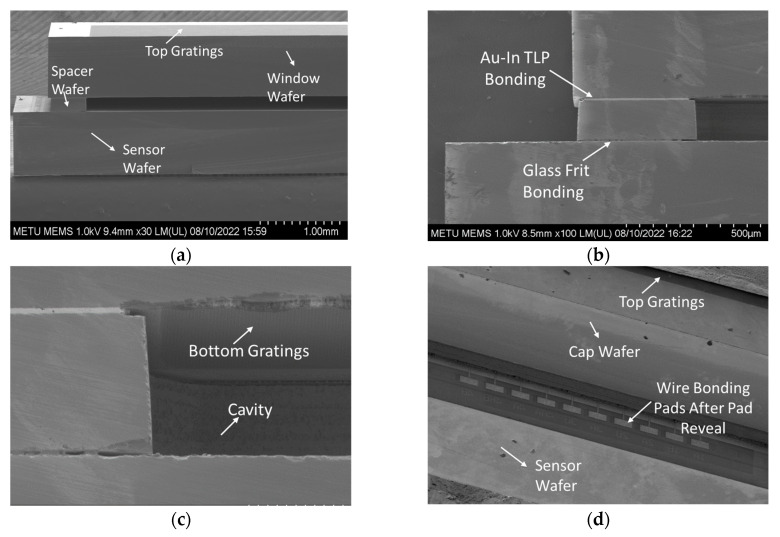
Cross-section SEM images showing: (**a**) the details of the encapsulated MEMS sensor; (**b**) Au-In TLP and glass frit bonding regions; (**c**) the cavity region and gratings (moth-eye structures) inside the cavity; (**d**) a perspective view of the bonding pad region of the sensor chip after cap wafer pad reveal.

**Figure 6 micromachines-15-00935-f006:**
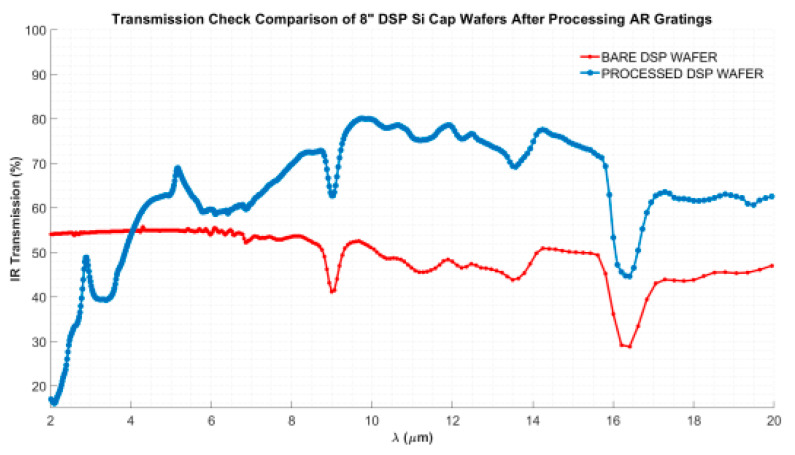
Infrared transmission measurement of the double-side moth-eye structures implemented in the window wafer (blue line) compared with the bare non-processed window wafer (red line) in the 2–20 μm wavelength region measured with the IR-VASE ellipsometer.

**Figure 7 micromachines-15-00935-f007:**
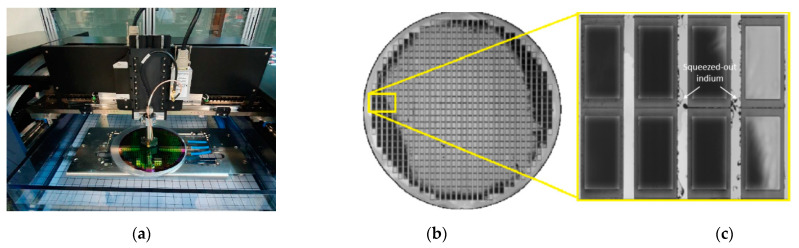
(**a**) The scanning acoustic microscope with the Au-In TLP bonded cap wafer stack on the chuck; (**b**) the SAM image of the bonded wafer; (**c**) a closer view of the multiple die region.

**Figure 8 micromachines-15-00935-f008:**
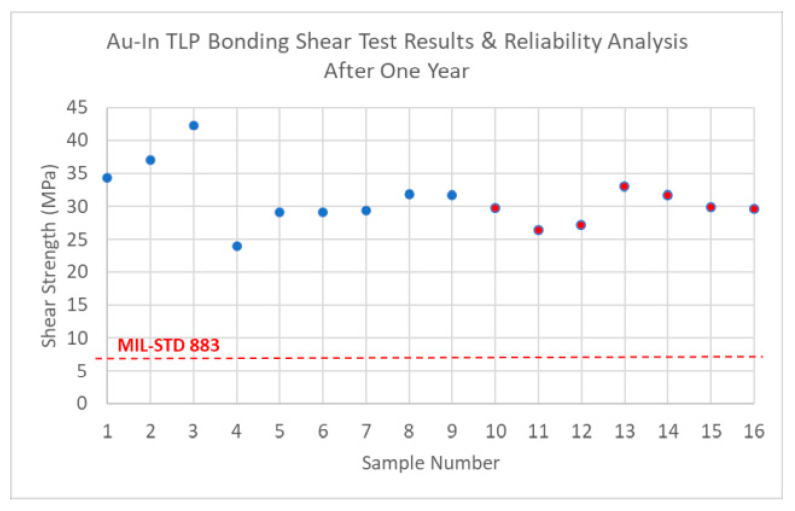
The shear test results of the Au-In TLP bonded samples. Sample measurements from 1 to 9 are the first measurements, shown in blue color dots, and from 10 to 16 are the measurements one year after the first measurement, shown in red color dots.

**Figure 9 micromachines-15-00935-f009:**
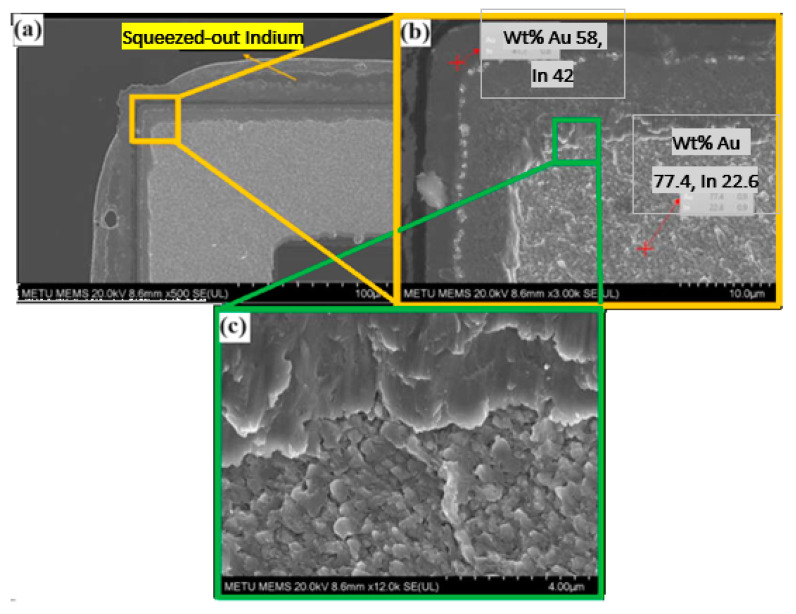
SEM images of the shear surface of Au-In TLP bonding: (**a**) the image of the Au-In TLP bonded stack diced and shear test applied case to show the squeeze out Indium; (**b**) a closer image also showing the EDS analyze results from the broken pieces’ Au-In alloy part; (**c**) a close-up SEM image of the shear surface of Au-In TLP bonding.

**Figure 10 micromachines-15-00935-f010:**
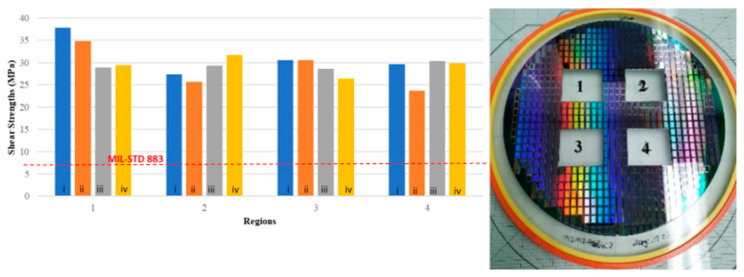
The comparison of shear test results of 4 regions. Graphic bars refer to (i) before annealing, (ii) annealing at 430 °C for 10 min, (iii) annealing at 430 °C for 30 min, and (iv) annealing at 430 °C for 60 min.

**Figure 11 micromachines-15-00935-f011:**
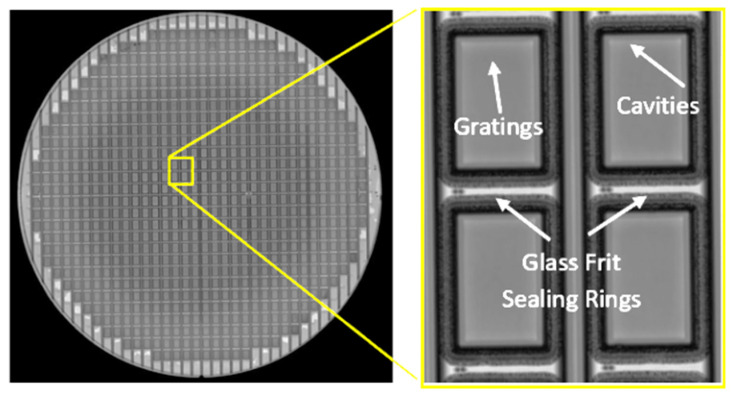
The SAM image of the bonded wafers for understanding the glass frit bonding quality.

**Figure 12 micromachines-15-00935-f012:**
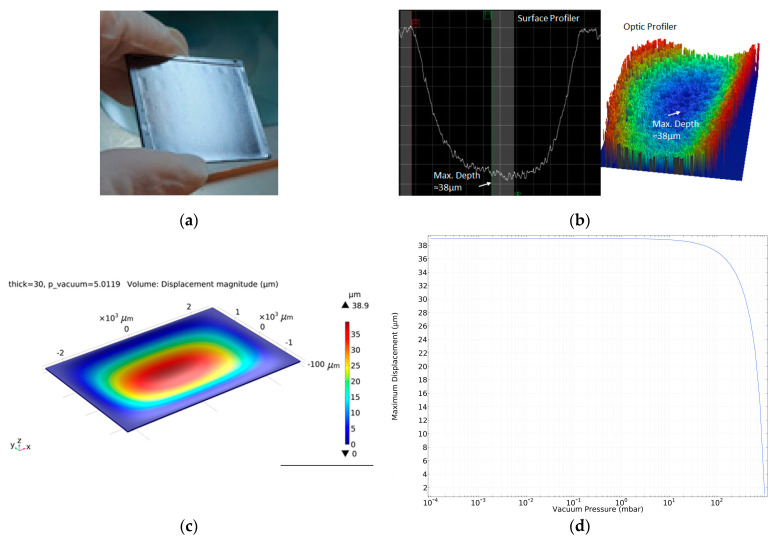
The images of: (**a**) bonded and diced piece including 5 × 7 dies thinned in DRIE to have a thin diaphragm over the vacuum cavity; (**b**) the one of the maximum deflected (≈38 µm) cap diaphragms measurement results in an optic and surface profiler; (**c**) the COMSOL simulation results showing that the maximum deflection can be ≈39 µm; (**d**) the maximum displacement versus vacuum pressure plot drawn according to COMSOL simulations; resulting in a measured vacuum level of approximately 5 mbar (500 Pa).

**Table 1 micromachines-15-00935-t001:** Summary of MEMS Wafer Level Packaging Methods.

Packaging Method	Process Temperature (°C)	Electric Characteristic	Topography Tolerance	Hermeticity
Thin Film Packaging [8,9,10,11]	250–1080	Insulating	Very Bad	Medium
Anodic Bonding [12,13]	250–450	Insulating	Bad	High
Plasma Activated Fusion Bonding [14,15,16]	200–400	Insulating	Very Bad	High
Polymer Adhesive Bonding [17,18]	<250	Insulating	Good	Low
Metal Thermo Comp. Bonding [19,20,21]	400–450	Conducting	Average	High
Solder/Eutectic Bonding [22,23,24,25,26,27,28,29,30]	~200 (Au-In)	Conducting	Good	High
	~300 (Au-Sn)			
	380–400 (Au-Si)			
Glass Frit Bonding [31,32,33,34,35,36,37]	430–450	Insulating	Very Good	High

**Table 2 micromachines-15-00935-t002:** The dimensions and mechanical properties of the cap diaphragm.

Parameters	Values
Si Poisson’s ratio ν	0.28
Si Young’s modulus, E (GPa)	170
Thickness (thinned, cap diaphragm), h (µm)	30
Length, l (µm)	5200
Width, w (µm)	3500

**Table 3 micromachines-15-00935-t003:** Summary of Pirani Gauge Designs and This Work.

Researcher	Gauge Type	Process Type	Pressure Range (Pa)
Shie et al. [47]	Cr/Pt resistor on a dielectric membrane	Bulk Micromachining	1.33 × 10^−5^–133
Stark et al. [48]	Cr/Pt resistor on a dielectric membrane	Surface Micromachining	0.13–1333
Chae et al. [49]	Cr/Pt resistor on a dielectric membrane anchored to Si	Surface Micromachining	2.67–266.65
Chae et al. [49]	p++ silicon coil microbridge	Dissolved wafer process	6.67–666.61
Mastrangelo and Muller [50]	Polysilicon microbridge	Surface Micromachining	9.99–9999.18
Stark et al. [51]	Polysilicon microbridge	Surface Micromachining	1.33–13,332.24
Mitchel et al. [52]	Polysilicon microbridge	Surface Micromachining	6.67–101,325
Topalli et al. [53]	p++ silicon coil microbridge	Dissolved wafer process	1.33–266.65
Topalli et al. [53]	100 µm thick Si coil microbridge	Silicon-on-glass process	2.67–266.65
This Work	Microbolometer-type resistive Pirani Vacuum Pixels	Surface Micromachining	0.13–533

## Data Availability

The original contributions presented in the study are included in the article, further inquiries can be directed to the corresponding author.

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
