# Peer review of "Wafer Level Vacuum Packaging of MEMS-Based Uncooled Infrared Sensors"

_micromachines, 2024, doi:10.3390/mi15080935_

Round 1
Reviewer 1 Report
Comments and Suggestions for Authors
In the manuscript “Wafer Level Vacuum Packaging of MEMS Based Uncooled Infrared Sensors” the authors have investigated wafer bonding Au/In SLID and glass frit for the supposed application of packaging of infrared sensors. Semiconductor wafer bonding technologies in general are interesting for various devices and packaging technologies, including infrared sensor packaging.
Detailed comments regarding the content of the manuscript:
Introduction
1) In the introduction section different wafer bonding technologies and packaging technologies are listed and compared in table 1. I advise to use the full names for each method for good style, e.g. anodic -> anodic bonding [method].
2) It seems the column “vacuum level” is either incorrect or mixes hermeticity and vacuum level. For example, the hermeticity of glass frit bonding is good, whereas its vacuum level is rather bad because of outgassing. In the case of anodic bonding, it is similar.
Experimental
1) Although in the previous section it is correctly stated, that the selection of the proper packaging technology is important, there is no explanation why Au/In SLID was selected for the lid fabrication.
2) For the bonding conditions information about the gas pressure during bonding is missing. It would help to evaluate the measured result.
3) There is no information about the bonding frame width. It would help to evaluate the result.
4) The thickness for Au and In do not fit the ratios needed to achieve the AuIn nor AuIn2 intermetallic phases. What was the rationale behind this?
5) There is no information about the glass frit type used. Judging from the bonding temperature probably a lead based type was used.
Results and discussion
1) It is explained that some indium was squeezed out during the bonding process. How it was determined it is indium? Does it mean the SLID process was not completed and intermetallic phases are not fully formed?
2) There are quite large deviations in the measured shear strength. Is there any explanation for it?
3) The annealing of the Au/In bonded samples was done at the same temperature as the glass frit bonding. If the already fully packaged stack (2 x bonding) was tested like that, there is not much meaning as it does not exceed the process temperature. If the mentioned annealing and test was done with only the lid and the purpose was to evaluate the influence of the glass frit bonding on the properties of the Au/In bond interface, it should be mentioned in the text.
4) Comment: A volume of 7.28mm3 is rather large for MEMS then small in my opinion.
5) There is information about the “best” vacuum level. Why there is no comprehensive description like for the shear strength?
6) For pressure measurements Pirani gauge and bolometer is mentioned. Which one actually was used?
Miscellaneous points:
1) Please use internationally accepted standard units, such as Pa in scientific papers. Especially mixing units for the same physical property is not good.
2) It is not immediately clear why references [4, 13, 18] of author T. Akin are included when basic technologies are referenced. There are more basic and more up-to-date references available. It gives a bit suspicion of the purpose of increasing own citations.
Author Response
First of all, we want to express our appreciation to the reviewer who examined the paper in detail and provided us with important feedback. We prepared the following responses and revised the manuscript as much as possible in order to satisfy the concerns of the reviewers. Please see the attachment. Our responses is attached in the word document which is named "author-coverletter-37632407.v1.docx

Reviewer 2 Report
Comments and Suggestions for Authors
The authors presented an approach to achieve wafer level vacuum packaging for MEMS-based uncooled infrared sensors. The manuscript is not acceptable at its present form as there are some issues that should be analyzed and addressed.
1. The abstract and conclusion sections mentioned the use of Pirani vacuum gauge for vacuum degree measurement, which is not well matched in the text. It is necessary to clearly show how Pirani vacuum gauge is designed and used in the test.
2. The authors chose different methods for vacuum degree measurement with or without getter. Why can't the same test method be used for comparison?
3. Comparisons on performance of packaged devices between this work and previous works are lacking.
4. The figures are of poor quality and need to be improved. For example:
a) as shown in Figure 1 and Figure 3, the text and color need to be optimized;
b) as shown in Figure 5, the subgraph sizes are different.
5. There are some problems in writing that should be corrected, such as: acronyms need to be indicated when they first appear, such as TCR.
Comments on the Quality of English LanguageThere are problems with writing specifications, which need to be modified and polished appropriately.
Author Response
First of all, we want to express our appreciation to the reviewer who examined the paper in detail and provided us with important feedback. We prepared the following responses and revised the manuscript as much as possible in order to satisfy the concerns of the reviewers. Please see the attachment.

Round 2
Reviewer 1 Report
Comments and Suggestions for Authors
Thank you for your answers and addressing my comments. Unfortunately, there are some remaining issues:
1) After your modification of table 1 it shows thin “Thin Film Bonding”. Thin film packaging is not a bonding method; it’s not related to bonding at all. I wonder whether you did read/understand the cited references?
3) Your added description is for sure correct. But in general it applies to all SLID bonding technologies. The interesting point is rather why you chose a low temperature material combination. After all you developed a meta structure surface to replace the temperature sensitive AR coating. There is e.g. Cu/Sn bonding with much cheaper materials. I probably should have given this precise point in my first comment.
8) Evidence of the occurrence of a liquid phase is the prove that the temperature was high enough for melting of indium and bonding was basically possible. It should not be used as sole evidence of bonding quality. If there is indeed elemental indium left after the bonding, it means the bonding process is not finished. It has a severe impact on reliability and therefore quality, as the strength of indium is much less than its intermetallic phases. It should not be left uncommented.
9) According to the text (line 257-259) and figure 9 the measured samples are from the same wafer. It escapes me how areas on the same wafer can suffer from differences in sample preparation and material property inhomogeneity. Could you please explain?
12) In the section about the strength measurement you gave information such as sample numbers, (basis) location and standard deviation. I was expecting a similar level of quality for the pressure measurements. (Or an explanation why it’s not given.)
13) I am still confused about your sensor structure. The technology and basic structure looks like a classical infrared sensing microbolometer. The described measurement indicates indeed a Pirani gauge. Does it mean your structure can also be used in some kind of heating mode, like a Pirani gauge? If so, this would be a good place to cite your previous work describing such structure.
14) Although it has been improved, the manuscript has still a mix of standard and non-standard units.
15) The suspicion about inappropriate self-citation comes up, as there is a rather large number of self-citations in this manuscript. If the citation is about previous work directly related to the manuscripts experiments and necessary or helpful for understanding (such us e.g. [44]), it is very welcome and can be in whatever number is needed. On the other hand, if there are many self-citations during introduction of general background it invites above mentioned suspicion. They may be justified, but please provide more than some general answer to my detailed comment. Adding more (own!) citations is not suitable to dispel this suspicion.
Could you please elaborate on the purpose of including table 3? According to the title, abstract and other text the manuscript is about infrared sensors.
Author Response
Firstly, we would like to express our appreciation to the reviewer who carefully examined the paper and provided us with valuable feedback once again. We have prepared the following responses and revised the manuscript extensively to address the remaining concerns raised by the reviewer. Please see the attachment.

Reviewer 2 Report
Comments and Suggestions for Authors
Dear authors,
Thanks for the changes you have added to the paper. From my point of view, this paper is now acceptable for publication.
Author Response
Dear Reviewer,
Thank you very much for your kind words and for finding the revised version acceptable for publication. We sincerely appreciate your guidance and feedback throughout this process.
Best regards,
Round 3
Reviewer 1 Report
Comments and Suggestions for Authors
Thank you for addressing the concerns listed. Some minor comments to your answers:
3) Eutectic bonding and SLID are in principle different. There is no intermetallic phase formation during eutectic bonding, an intermixed crystal lattice of the constituting elements is formed instead. The only common point is the occurrence of a liquid phase during the process.
Some material combinations (e.g. Au/Sn) can be used for either process (at different compositions).
4) According to your EDX measurements: what was squeezed out is rather a liquid phase of Au and In, than just In. Therefore, a reliability issue is unlikely.